# Biological Potential of Alternative Kombucha Beverages Fermented on Essential Oil Distillation By-Products

**Aleksandra Ranitović** [1,*], **Olja Šovljanski** [1,*], **Milica Aćimović** [2], **Lato Pezo** [3], **Ana Tomić** [1], **Vanja Travičić** [1], **Anja Saveljić** [1], **Dragoljub Cvetković** [1], **Gordana Ćetković** [1], **Jelena Vulić** [1] and **Siniša Markov** [1]

1   Faculty of Technology Novi Sad, University of Novi Sad, Bulevar Cara Lazara 1, 21000 Novi Sad, Serbia
2   Institute of Field and Vegetable Crops Novi Sad, Maksima Gorkog 30, 21000 Novi Sad, Serbia
3   Institute of General and Physical Chemistry, Studentski Trg 12-16, 11000 Belgrade, Serbia
*   Correspondence: a.ranitovic@uns.ac.rs (A.R.); oljasovljanski@uns.ac.rs (O.Š.)

**Abstract:** The complete waste streams (solid waste residue, wastewater, and hydrolate) from the essential oil production of basil, chamomile, lavender, rosemary, and hyssop plants were used as a cultivation media for fermentations of a health-beneficial beverage called kombucha. Considering that these waste streams have not been used as a medium for obtaining kombucha, the main focus of this study was on the biological profiling and sensory analysis of newly-obtained kombucha beverages. According to fermentation parameters and advanced mathematical modelling, it can be concluded that kombucha made from chamomile essential oil by-products achieved the fastest successful kombucha fermentation, with a maximal titratable acidity of 7.2 g/L and a minimal pH value of 2.8. The results of other kombucha fermentations varied between the chosen plant and the waste stream used for beverage production. The obtained phenol and flavonoid contents were in the range of 12.4–56.46 mg GA/100 mL and 0.25–5.07 mg RU/100 mL, respectively. Higher antioxidant capacity as well as anti-inflammatory and antihyperglycemic activities of all kombucha beverages were observed compared to controls. Briefly, achieved DPPH, ABTS, and reducing power values were in the range 30.28–73.70, 192.25–683.29, and 19.37–82.76 mmol TE/100 mL, respectively. According to sensory analysis, the best performance or complete acceptability was noted for kombucha beverages made from lavender and hyssops (in the case of solid waste stream mixed with hydrolate) as well as basil (in the case of concentrated wastewater and hydrolate).

**Keywords:** kombucha; alternative crops; antioxidant activity; antihyperglycemic activity; anti-inflammatory activity; fermented beverage; sensory analysis; flavonoids; phenols

## 1. Introduction

Solid waste residue, wastewater, and hydrolate are by-products (waste streams) generated during the steam distillation of essential oils [1]. In light of the circular economy, which is promoted worldwide, researchers from all scientific fields are focused on recovering waste material for value-added products [2]. Hydrolates are highly studied, and they have a wide range of documented bioactivities, such as antimicrobial, antioxidant, and anti-inflammatory, as well as applications in the food industry (including preservation of fruits and vegetables and flavoring sweets, beverages, and soft drinks), aromatherapy (including perfumes and air fresheners), cosmetics (including creams and soaps), and organic agriculture (including the biological control of weeds, diseases, and pests) [3]. Solid waste residue (essential oil-exhausted biomass) has a modification of the plant structure during essential oil extraction, but it could still be valorized as a source of antioxidants in the food and feed, cosmetic, and pharmaceutical industries, as well as a source of biomass for composting in organic agricultural production [4–6]. Further, investigations show that wastewater (deodorized water extract) also possesses significant antioxidant properties,

mainly due to the polyphenols which remain in this by-product [7–10]. The waste stream in the production of essential oil is actually tea prepared from certain plant(s), and it is rich in nutrients and bioactive compounds which have passed from plant material to water during the boiling.

The global kombucha market is currently thriving as functional foods and beverages are getting more attention and people's consciousness about consuming healthier nutritional options is expanding. The market is expected to reach USD 3.5 to 5 billion by 2025 [11]. Day by day, there are more reports of innovative fermented products being investigated using raw materials other than tea, such as fruit or vegetable juices, herbal or plant infusions, milk, and food industry by-products. A vast variety of substrates used in the production of kombucha makes beverages with different chemical compositions and types of biological activity. Due to an ever-present need for the valorization of industry by-products, the literature is filled with different by-products and wastes that can be utilized for kombucha production [12]. For example, Jayabalan et al. [13] noted that tea solid waste could be used as a raw material to produce fermented beverages, such as kombucha. Some of the investigations conducted on the use of molasses as a substrate for the fermentation process have shown that it is attractive due to its low price and the presence of various components [14]. Also, Belloso Morales and Hernandez- Sanchez [15] successfully cultivated a type of fermented beverage with cheese whey, which is a by-product of milk production [16]. Interestingly, research by Pure and Pure [17] showed that banana peel, as a waste plant material, is a good substrate for the preparation of fermented kombucha beverages, with a new taste and color and significant phenolic content and antioxidant activity. Beverages obtained from soybean whey, which is a by-product of soy processing, had fruity and floral flavors, showing potential for kombucha production [18]. Coffee pulp, the main waste stream of wet coffee processing, has also found its use as a substrate for kombucha production in the research of Muzaifa [19], who used raw materials from cascara Gayo arabica coffee. Valorization of winery effluent for kombucha production was investigated by Vukmanović et al. [20], and the obtained beverage was found to be appropriate for consuming, having a wine-like note.

Even though there are plenty of studies regarding the use of wastes and by-products in kombucha production, there were no examples of using waste streams from essential oil production prior to this study. Therefore, this investigation aimed to recover all by-products obtained during essential oil distillation (solid waste residue, wastewater, and hydrolate) from commonly known and widely cultivated plants, such as basil, chamomile, lavender, rosemary, and hyssop, and used the by-products as a cultivation media for the fermentation of a functional beverage called kombucha. The main focus of this study is the biological profiling and sensory analysis of newly-obtained kombucha beverages based on alternative fermentation substrates.

## 2. Materials and Methods

### 2.1. Essential Oil Production and Waste Streams Obtaining

Basil (*Ocimum basilicum* L.) cv 'Sitnolisni', chamomile (*Matricaria chamomilla* L.) cv 'Tetraploidna', lavender (*Lavandula angustifolia* Mill.) cv 'Primorska', rosemary (*Rosmarinus officinalis* L.) cv 'Plavi', and hyssop (*Hyssopus officinalis* L.) cv 'Domaći ljubičasti' were grown at the Institute of Field and Vegetable Crops Novi Sad (IFVCNS) during 2021. Plants were harvested at the optimal time for each plant species, dried in a solar dryer, and then submitted to steam distillation. Steam distillation was performed in a small-scale distillation unit (Inox, Bački Petrovac, Serbia) at the IFVCNS. This process was previously described in detail in Aćimović et al. [21]. Briefly, stainless steel distillation vessels were filled with dry plant material, enclosed with stainless steel lids, and supplied with steam. The duration of the distillation process was different for each investigated plant species according to European Pharmacopoeia [22]. The essential oil and the hydrolate were separated in a Florentine flask, while solid waste residue and wastewater remained in the distillation vessel. When the distillation process was completed, the boiling wastewater

was sluiced from the vessel through the stainless-steel grid (5 mm), and the solid waste residue was dumped via a servomechanism system.

### 2.2. Kombucha Fermentation

Fermentation was performed by using the local tea fungus culture, which is traditionally used for all research at the Laboratory of Microbiology at the Faculty of Technology Novi Sad. The authors have previously reported that the fungus culture contains at least *Saccharomycodes ludwigii*, *S. cerevisiae*, *S. bisporus*, *Torulopsis* spp., *Zygosaccharomyces* spp., and two bacteria of the *Acetobacter* genera [23].

Waste streams from essential oil production were used to create a cultivation medium, and three different batches were made (batch A, batch B, and batch C). Differences between batches were based on the used waste materials. Namely, batches A and B were made using waste plant material in different concentrations. Briefly, 3 or 9 g/L of waste plant material was added to the boiled and sweetened (70 g/L sucrose) hydrolate of the same plant (which is also a by-product of essential oil hydrodistillation) and removed after a quarter of an hour by filtration. The third batch C was made from concentrated wastewater which was diluted with hydrolate in a ratio of 1:10 and sweetened (70 g/L sucrose) before the boiling process. For all three batches, after reaching room temperature, the obtained cultivation medium was inoculated with 10% (*v*/*v*) of the fermentation broth from the previous fermentation (the kombucha obtained as traditional kombucha from black tea). The cultivation medium was transferred into glass vessels, and sterile gauze was added to the vessel to prevent contamination during cultivation at 28 °C.

The fermentation of all batches was monitored (0, 2, 5, 7, and 9 days) using two essential chemical parameters: pH value and titratable acidity (TA). The process was stopped after reaching the optimal acid content. Namely, in order to obtain a pleasantly sour beverage, fermentation should be stopped after achieving a pH value of 3–4 and a TA of fermentation broth between 4 and 4.5 g/L, which has been validated by kombucha consumers [24]. Furthermore, yeasts and acetic acid bacteria (AAB) numbers, as microbiological parameters, followed the same fermentation time as the chemical parameters. Microbiological profiles were determined by a plate pour method using Sabouraud-4% Maltose Agar (HiMedia, Mumbai, India) with the addition of 50 mg/L of chloramphenicol (SigmaAldrich, St. Louis, MI, USA) and Yeast Peptone Mannitol Agar (Difco, Detroit, MI, USA), which contained 500 mg/L cycloheximide (SigmaAldrich, St. Louis, USA) for yeast and ABB, respectively. The plates for yeast determination were incubated for 72 hours at 28 °C, while the incubation for ABB was 5–7 days.

If batches did not reach the optimal value after 9 days of fermentation, the process was stopped, and these batches were defined as unsuccessful and not selected for further steps. The pH value was determined by using a pH meter (HI 99181, HANNA Instruments, Woonsocket, RI, USA), while titratable acidity was determined using 0.1 M NaOH solution. Briefly, after removing $CO_2$ from the final kombucha samples, an aliquot was titrated with the previously mentioned base. The titratable acidity was expressed as grams of acetic acid per liter in the sample.

### 2.3. Antioxidant Capacity

The total polyphenolic contents for the kombucha samples were determined spectrophotometrically using the microscale-adapted Folin-Ciocalteau method [25]. In short, 15 µL of extract, 170 µL of distilled water, 12 µL of the Folin-Ciocalteu's reagent (2 M), and 30 µL of 20% sodium carbonate were used to prepare the reaction mixture that was then put in 96 well microplates. The absorbance was measured at 750 nm after 1 hour using distilled water as a blank. The obtained results were expressed as gallic acid equivalents (GAE) per 100 g sample.

The aluminum chloride colorimetric assay [26] was adapted for a 96-well microplate and used to spectrophotometrically determine the content of the flavonoids found in different kombucha samples. The probes were prepared in microplate wells by mixing

25 μL of kombucha extracts (or solvent for the blank probe), 100 μL of distilled water, and 7.5 μL of 5% sodium nitrate. Then, after 5 min, 7.5% of aluminum-chloride solution (10%) was added, and then after 6 min, 50 μL of 1 M sodium hydroxide and 100 μL distilled water were added. The absorbance of the prepared reaction mixture was measured at 510 nm. The results were expressed as rutin equivalents (RE) per 100 g sample.

Three antioxidant tests were done: 2,2-diphenyl-1-picrylhydrazyl (DPPH), 2,2′-azino-bis-3-ethylbenzothiazoline-6-sulphonic acid (ABTS), and reducing power (RP).

The antioxidant capacity of the samples was determined by the ability of the samples to scavenge 2,2-diphenyl-1-picrylhydrazyl (DPPH) radical with a spectrophotometric assay reported by Mena et al. [27]. In brief, a 250 μl of DPPH solution in methanol (0.89 mM) was mixed with 10 μL of the sample in a microplate well and left in the dark at room temperature. Absorbances were determined at 515 nm after 50 min. Methanol was used as a blank. The DPPH activity was calculated using Equation (1)

$$\text{DPPH (\%)} = ((A_{control} - A_{sample})/A_{control}) \times 100 \tag{1}$$

where the $A_{control}$ is the absorbance of the control and $A_{sample}$ is the absorbance of the sample. Results were also expressed as μmol Trolox equivalents (TE)/100 g sample dry weight.

The ABTS•+ radical scavenging assay of the kombucha samples was valued using the method described by Aborus et al. [28]. Firstly, 250 μL of activated ABTS•+ (with $MnO_2$) was added to the microplate well, and the initial absorbance was measured. Then, 2 μL of each sample was added to the plate and left to incubate at 25 °C for 35 min before the final absorbance was measured. Both absorbances were measured at 414 nm, and water was used as a blank.

Using the method described by Oyaizu [29], reduction power (RP) was determined, which is based on monitoring the reduction capacity of the subject sample for $Fe^{3+} \rightarrow Fe^{2+}$ transformation. The volumes of 75 μL of sample solution or 75 μL of extractant (blank), 75 μL of sodium phosphate buffer pH 6.6, and 75 μL of 1% m/V potassium ferricyanide were mixed. Solutions were tempered in an aqueous bath for 20 min at 50 °C. They were cooled, and then 75 μL of 10% m/V was added to the solution of trichloroacetic acid. Next, the solutions were centrifuged at 3000 rpm for 10 min. After centrifugation, 50 μL of distilled water and 10 μL were added to 50 μL of carefully separated supernatant 0.1% *w/v* ferric chloride. The absorbances were measured immediately at a wavelength of 700 nm.

The Trolox was used as a standard antioxidant, and the results of the antioxidant activities were expressed as μmol Trolox equivalent (TE) per 100 g of sample (μmol TE/100 g).

### 2.4. Pharmacological Activities

### 2.4.1. Anti-Inflammatory Activity

In vitro evaluation of the anti-inflammatory characteristics of the samples was carried out with the test protein denaturation. The reaction mixture (5 mL) consisted of 0.2 ml of egg albumin (from a fresh chicken egg), 2.8 mL of phosphate buffer pH 6.4, and 2 mL of sample extract. A similar volume (2 mL) of distilled water served as a control. The mixtures were then incubated at 37 ± 2 °C in a Buchi incubator (Grejno bath B-491, Switzerland) for 15 min, and then heated to 70 °C for 5 min. After cooling, their absorbance was measured at 660 nm on a Multiscan GO microtiter plate reader (Thermo Fisher Scientific Inc., Waltham, MA, USA). The percentage inhibition of protein denaturation (AIA) was calculated based on Equation (2)

$$\text{AIA (\%)} = (A_{control} - A_{Sample})/A_{control} \times 100 \tag{2}$$

where $A_{control}$ and $A_{sample}$ are the absorbances of the reaction mixture of the control and the sample. Diclofenac sodium in the concentration of 20 mg/mL was used as a reference drug and treated the same for the determination of absorption and anti-inflammatory activities.

### 2.4.2. Antihyperglycemic Assay

The inhibition potential of $\alpha$-glucosidase (AHgA) was determined spectrophotometrically according to the method described by Tumbas Šaponjac et al. [30]. The necessary solutions and reagents for this test were 10 mM potassium phosphate buffer pH 7, 2 mM substrate solution (4-nitrophenyl $\alpha$-D-glucopyranoside) in the buffer, samples dissolved in the buffer, and enzyme ($\alpha$-glucosidase) dissolved in the buffer. The preparation of enzymes involved dissolving 1.35 mg of $\alpha$-glucosidase in 1 mL of buffer, after which it was measured aliquot of 40 μL in a 10 mL dish and made up to the mark. Reaction mixtures were prepared in the openings of microtiter plates in the following way: 100 μL substrate was mixed with 20 μL of sample and 100 μL of enzyme solution. The absorbance of 4-nitrophenol of 4-nitrophenyl-$\alpha$-D-glucopyranoside was measured at 405 nm before and after incubation of 10 min at 37 °C, and it was compared to the control absorbance to calculate the potential inhibitions. AHgA (%) was calculated based on Equation (3)

$$AHgA\ (\%) = (\Delta A_{control} - \Delta A_{Sample}) / \Delta A_{control} \times 100 \tag{3}$$

where $\Delta A_{control}$ and $\Delta A_{Sample}$ are the different absorbances of the reaction mixture of the control and the sample before exposure to the enzyme and after 10 minutes of incubation with the enzyme. A control blank consisted of a 100 μL substrate and 120 μL buffer, a 100 μL control substrate, a 20 μL buffer and 100 μL enzyme, and a sample blank of 100 μL substrate, 20 μL sample, and 100 μL buffer.

### 2.5. Sensory Analysis

The sensory analysis of the final kombucha beverages was determined using a trained sensory panel (3 males and 7 females, 24 to 64 years old). The structure of panelists followed the latest research which indicates that consumers of kombucha beverages are mostly young females [31]. All panelists were asked to sign a consent form that included a list of potential allergens that may have been in the samples. Sensory descriptors for six different categories, namely smell, taste, acidity, sweetness, color tone, and overall acceptability, were evaluated. For each category, a descriptive evaluation was made.

- Smell: vinegar mild (1); vinegar strong (2); yeast (3); plant (4); other (5).
- Taste: mild vinegar (1); vinegar strong (2); yeast (3); plant (4); other (5).
- Acidity: imperceptibly acidic (1); slightly sour (2); medium sour (3); noticeably sour (4); too sour (5).
- Sweetness: imperceptibly sweet (1); slightly sweet (2); average sweet (3); noticeably sweet (4); too sweet (5).
- Color tone: pale (transparent) (1); light yellow (2); golden-yellow (3); light brown (4); dark brown (5).
- Overall acceptability: completely unacceptable (1); partially unacceptable (2); indifferent (3); partially acceptable (4); completely acceptable (5).

The sensory analysis was performed at a temperature of 22 °C, and the kombucha beverages were refrigerated before consummation at 4 °C. The samples were presented in glass cups coded with two-digit random numbers and were evaluated in one consecutive session.

### 2.6. Statistical Analysis

In this investigation, microbial growth (yeasts and acetic acid bacteria), pH value, and titratable acidity kinetics modelling were implemented, applying the four-parameter sigmoidal computational model, which is convenient for biological systems and described in detail by Romano et al. [32]. The projected data should be shaped to the S-shaped curve, while the model could be written in the form of Equation (4).

$$y(t) = d + \frac{a - d}{1 + \left(\frac{t}{c}\right)^b} \tag{4}$$

The yeast isolate number and acetic acid bacteria number (log CFU/mL) during incubation time (hours) were titled as $y(t)$, while the regression coefficients could be explained as follows: $a$—the minimum of the experimentally gained values (at $t = 0$), $d$—the maximally acquired value (at $t = \infty$), $c$—the inflection point (the point between $a$ and $d$), and $b$—the Hill's slope (the steepness of the inflection point $c$). The same applied to pH values (/) and titratable acidity (g/L) values.

The intensities of the sample attributes for sensory analysis were assessed based on the 1 to 5 linear Likert scale. The sensory evaluation was performed using a balanced factorial design. The sample evaluation was specified by the Experiment design for sensory analysis with XLSTAT-MX (XLSTAT 2018.7. Addinsoft. http://www.xlstat.com, accessed on 23 August 2022). Each assessor received 20 mL of each sample which was supplied separately on a white plastic plate and coded with three randomly selected numbers. The samples were investigated in duplicate in individual air-conditioned (22 °C) sensory booths. Room-temperature water was used for palate cleansing.

## 3. Results and Discussion

### 3.1. Kombucha Fermentation

Kombucha represents a fermented beverage in which acid-tolerant species of a symbiotic culture of bacteria and yeasts are used. As two main substrates for kombucha fermentation, black tea and sugar are traditionally used, but recently, many modifications have occurred to improve the quality and the sensory acceptability of the final drink, and most have been directed at using alternative substrates for fermentation [33]. In this study, kombucha fermentation was modified, and instead of using traditional black tea as the substrate, some alternatives were involved. Briefly, three batches (A, B, and C) were prepared based on the different waste materials of the essential oil distillation process according to the procedure previously explained (see Section 2.2). The total number of fermentations amounted to 15.

The fermentation process was monitored at 0, 2, 5, 7, and 9 days of fermentation which determined the four parameters: the yeasts number, the acetic acid bacteria (AAB) number, the pH value, and the titratable acidity (TA). The process was stopped after reaching the optimal content of acid (pH values of 3–4 and TA 4–4.5 g/L). The obtained results were evaluated using a kinetics modelling approach that is suitable for biological processes, such as kombucha fermentation [32]. According to Equation (4), the kinetics evaluation of the fermentation was performed using the averaged values of three batches (A, B, and C) for basil, chamomile, lavender, rosemary, and hyssop to present differences in the fermentation process depending on the plant. The regression coefficients were calculated for the averaged values of the yeast number, the AAB number, the pH values, and the TA values kinetics models. The averaged experimental values and kinetics models are presented in Figure 1.

According to the results presented in Figure 1a,b, it is obvious that in almost all tested samples, the number of yeasts and the AAB reaches its maximum after the second day of fermentation, which is in accordance with previously published research [34]. The initial number of both yeasts and AAB approximately increases for one log CFU/mL within the first two days. The only exception is noticed for the kombucha beverages prepared with basil, rosemary, and hyssop, where the initial number of yeasts remains the same until the end of the fermentation process. Besides the microbiological parameters, in the same time interval, the pH and TA values were tracked for all kombucha fermentation (Figure 1c,d).

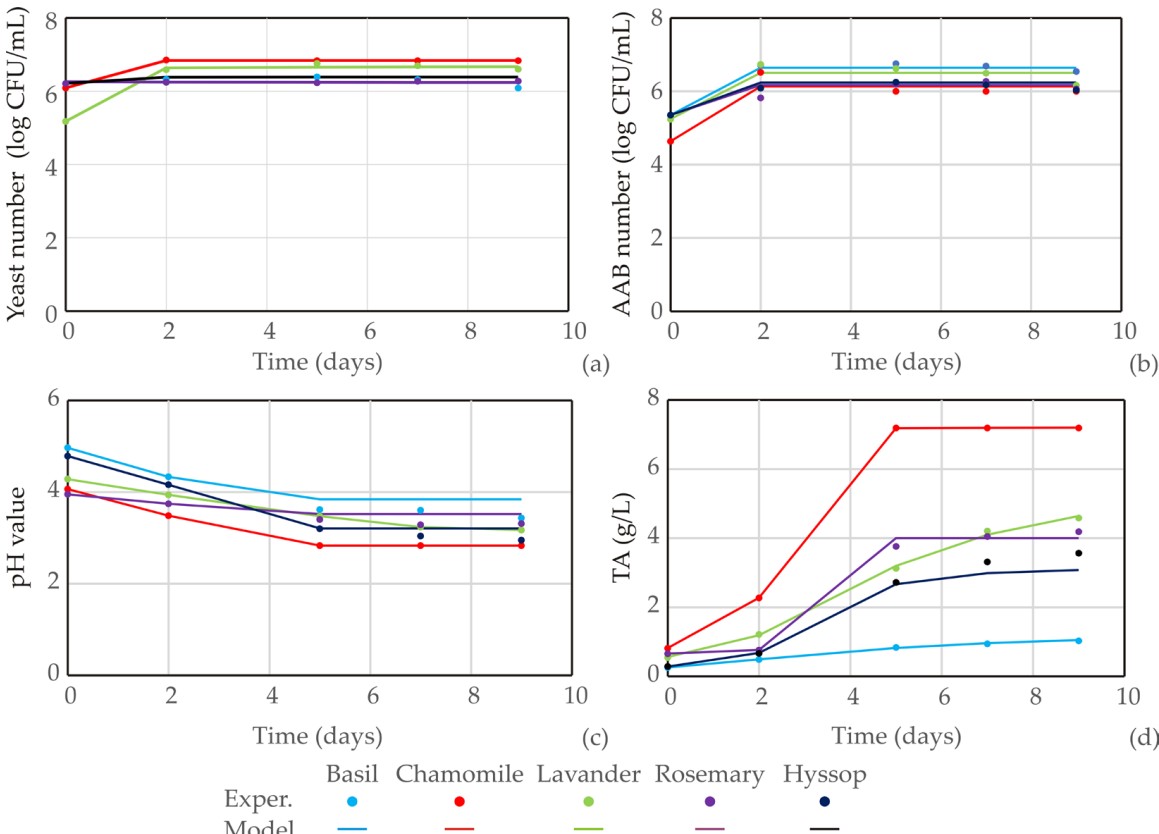

**Figure 1.** Experimental values (points) and kinetics curves (lines) for: (**a**) yeast number, (**b**) AAB number, (**c**) pH values, and (**d**) TA value during kombucha fermentation.

Based on the obtained results, it can be seen that the pH values of all kombucha beverages gradually lower until the fifth day of fermentation, while the TA values gets progressively higher. Namely, for obtaining a pleasantly sour beverage, fermentation should be stopped after achieving a pH value between 3 and 4 and TA value between 4 and 4.5 g/L, which is validated by kombucha consumers [24]. After the fifth day, they reach pH and TA values that remain equal until the ninth day of fermentation. The phenomenon of reducing the pH value and increasing the TA value at the same time during kombucha fermentation is common and usually expected [34]. The observed changes in the pH and TA values were slower only in the case of the lavender-based kombucha. Also, the maximum TA value of the kombucha with basil was significantly low (approx. 1 g/L), and after the fermentation process, these kombucha beverages had the highest pH value (3.9 units), which indicates a slow and incomplete fermentation process. On the other hand, the maximum TA value (7.2 g/L) and the minimum pH value (2.8 units) were observed for the kombucha with chamomile.

Table 1 summarizes the regression coefficients (*a*, *b*, *c*, and *d*) of the observed sigmoidal mathematical models of the kombucha fermentation processes which explain the trends (the speed and the intensity) of the investigated kinetics curves for the kombucha fermentation parameters, including: the yeast number, the AAB number, the pH values, and the TA values. The *p*-value for each term in the regression models was tested for the null hypothesis that the regression coefficient is equal to zero (i.e., it produces no effect on the tested variable). The regression function of reduction of all regression functions was found statistically significant at $p < 0.05$ level.

**Table 1.** Regression coefficients for kinetic models for kombucha fermentation based on Equation (4).

| | | Plant | | | | |
|---|---|---|---|---|---|---|
| | | Basil | Chamomile | Lavender | Rosemary | Hyssop |
| | | Kombucha fermentation parameters | | | | |
| | | Yeast number (log CFU/mL) | | | | |
| Regression parameters | *d* | 6.232 | 6.837 | 7.975 | 6.232 | 6.384 |
| | *a* | 6.257 | 6.085 | 5.175 | 6.257 | 6.210 |
| | *c* | 1.911 | 0.100 | 0.100 | 1.911 | 0.100 |
| | *b* | 0.759 | 4.509 | 0.029 | 0.759 | 1.280 |
| | | AAB number (log CFU/mL) | | | | |
| Regression parameters | *d* | 6.642 | 6.131 | 6.503 | 6.171 | 6.234 |
| | *a* | 5.350 | 4.635 | 5.240 | 5.350 | 5.350 |
| | *c* | 0.100 | 0.100 | 0.100 | 0.100 | 0.100 |
| | *b* | 5.225 | 5.225 | 5.225 | 5.225 | 5.225 |
| | | pH value | | | | |
| Regression parameters | *d* | 3.840 | 2.830 | 3.354 | 3.519 | 3.207 |
| | *a* | 4.965 | 4.063 | 4.280 | 3.950 | 4.783 |
| | *c* | 1.963 | 2.018 | 2.085 | 2.010 | 2.063 |
| | *b* | 13.431 | 13.374 | 12.584 | 13.395 | 13.472 |
| | | TA (g/L) | | | | |
| Regression parameters | *d* | 1.405 | 7.193 | 5.696 | 3.999 | 3.141 |
| | *a* | 0.257 | 0.815 | 0.561 | 0.658 | 0.281 |
| | *c* | 5.121 | 2.319 | 4.882 | 2.408 | 3.252 |
| | *b* | 1.443 | 8.206 | 2.207 | 18.508 | 3.743 |

　　　The presented four-parameter sigmoidal mathematical models appear to be simple, robust, and accurate (all coefficients of determination are greater than 0.775, according to Table 2). The quality of the model fit was tested, and the residual analysis of the developed model is presented in Table 2. The mathematical models have an insignificant lack of fit tests which means that all the models represent the data satisfactorily. A high $r^2$ shows that the variation is accounted for and that the data fit satisfactorily in the proposed model. It can be concluded that evaluated kinetics models can be used in the following studies as part of the monitoring of kombucha fermentation and the determination of the main fermentation parameters for obtaining the final product.

　　　After the evaluation of the kombucha fermentation process, the selection of the finished kombucha was done based on the dependence of the achieved pH and TA values. The typical fermentation process is done for a maximum of 7 to 10 days at room temperature (18–30 °C) [33,34], which is enough time to obtain a beverage with good physicochemical, microbiological, and sensory qualities. In another words, if the final kombucha product is not reached for 10 days, the fermentation process is determined as unfinished and insufficiently efficient for a potential *scale-up* and industrial process. Samples that achieved the desired pH and TA values for the fermentation time include the following kombucha fermentation: Basil batch C, Chamomile batch A, Chamomile batch B, Chamomile batch C, Rosemary batch C, Lavender batch B, Lavender batch C, Hyssop batch B, and Hyssop batch C.

　　　According to the obtained results (Tables 1 and 2), it is obvious that only kombucha prepared with Chamomile achieved the desired pH and TA values within the expected fermentation time for all three batches A, B, and C. This is in good correlation with the results presented in Figure 1, where it is clearly presented that the fermentation of the kombucha based on chamomile is the fastest considering all tracked microbiological and technological parameters. The success of fermentation can be defined through the applied batches: batch C > batch B > batch A. Namely, all fermentations carried out in batch

C can be defined as successful since the pH and TA values reached the desired level within the defined fermentation time. The second most successful batch is batch B in the case of chamomile, lavender, and hyssop, while batch A achieved a satisfactory level of technological parameters only when the chamomile was used.

**Table 2.** The "goodness of fit" kinetics models.

| | $\chi^2$ | RMSE | MBE | MPE | $r^2$ | Skew | Kurt | Mean | StDev | Var |
|---|---|---|---|---|---|---|---|---|---|---|
| | | | | Yeast number | | | | | | |
| Basil | 0.015 | 0.110 | 0.023 | 1.629 | 0.998 | −0.703 | −0.773 | 0.023 | 0.121 | 0.015 |
| Chamomile | 0.000 | 0.008 | 0.000 | 0.088 | 0.999 | 1.926 | 3.672 | 0.000 | 0.009 | 0.000 |
| Lavender | 0.005 | 0.061 | 0.006 | 0.749 | 0.990 | 0.720 | −0.276 | 0.006 | 0.068 | 0.005 |
| Rosemary | 0.001 | 0.031 | 0.002 | 0.423 | 0.775 | −0.454 | −0.755 | 0.002 | 0.035 | 0.001 |
| Hyssop | 0.042 | 0.182 | 0.069 | 2.135 | 0.885 | 0.876 | −0.880 | 0.069 | 0.189 | 0.036 |
| | | | | AAB number | | | | | | |
| Basil | 0.007 | 0.077 | 0.024 | 0.988 | 0.982 | −1.085 | 1.096 | 0.024 | 0.082 | 0.007 |
| Chamomile | 0.050 | 0.200 | 0.000 | 2.480 | 0.899 | 1.925 | 3.667 | 0.000 | 0.224 | 0.050 |
| Lavender | 0.047 | 0.195 | 0.028 | 2.335 | 0.874 | −0.914 | 1.318 | 0.028 | 0.215 | 0.046 |
| Rosemary | 0.036 | 0.170 | −0.067 | 1.949 | 0.789 | −1.352 | 1.788 | −0.067 | 0.175 | 0.031 |
| Hyssop | 0.017 | 0.116 | −0.080 | 1.395 | 0.948 | −0.428 | −2.228 | −0.080 | 0.094 | 0.009 |
| pH value | | | | | | | | | | |
| Basil | 0.069 | 0.235 | −0.175 | 4.977 | 0.975 | −0.185 | −1.530 | −0.175 | 0.175 | 0.031 |
| Chamomile | 0.000 | 0.000 | 0.000 | 0.000 | 1.000 | −1.904 | 3.789 | 0.000 | 0.000 | 0.000 |
| Lavender | 0.016 | 0.112 | −0.038 | 2.605 | 0.944 | 0.003 | −0.862 | −0.038 | 0.118 | 0.014 |
| Rosemary | 0.029 | 0.151 | −0.113 | 3.417 | 0.961 | 0.051 | −2.872 | −0.113 | 0.112 | 0.013 |
| Hyssop | 0.024 | 0.138 | −0.088 | 2.934 | 0.987 | −0.913 | −1.549 | −0.088 | 0.120 | 0.014 |
| TA value | | | | | | | | | | |
| Basil | 0.000 | 0.014 | −0.004 | 1.427 | 0.998 | 0.682 | −0.078 | −0.004 | 0.015 | 0.000 |
| Chamomile | 0.000 | 0.004 | 0.000 | 0.083 | 1.000 | 1.905 | 3.636 | 0.000 | 0.005 | 0.000 |
| Lavender | 0.005 | 0.064 | 0.000 | 2.140 | 0.998 | 0.910 | 0.192 | 0.000 | 0.072 | 0.005 |
| Rosemary | 0.024 | 0.137 | 0.000 | 2.409 | 0.993 | −0.781 | 2.000 | 0.000 | 0.153 | 0.024 |
| Hyssop | 0.087 | 0.264 | 0.173 | 6.087 | 0.992 | 0.849 | −1.596 | 0.173 | 0.223 | 0.050 |

$\chi^2$, reduced chi-square; RMSE, root mean square error; MBE, mean bias error; $r^2$, coefficient of determination; Skew, skewness; Kurt, kurtosis; Mean, mean of the residuals; StDev, standard deviation of the residuals; Var, variance of the residuals.

The gained differences in fermentation can be based on the background of the used waste plant material from the production of essential oils, which vary. To be specific, batch C refers to the wastewater obtained after boiling the plant material as a production step in essential oil preparation. As demonstrated in this study, a rich wastewater stream from the essential oil production of all tested plants is definitely an excellent base for kombucha preparation. On the other hand, batches A and B were prepared on solid waste plant residue, just with a different amount (3 g/L and 9 g/L, respectively). In line with the results presented in Tables 1 and 2, it can be concluded that the reuse of solid waste residues depends on the used plant. In the re-used waste plant materials, the amount of nutrients and active compounds necessary for kombucha production is considerably lower than in fresh plant materials. Therefore, if the initial amount of solid waste material is low, such as in batch A, there is a threat that the kombucha fermentation will be unfinished or insufficient. In most cases, when increasing the amount of solid waste plant residue as a base for kombucha preparation, the amount of necessary nutrients and active compounds increases together with the success of the kombucha fermentation. Besides the initial amount of solid waste residue used for kombucha preparation, the retrieving of a product with good physicochemical, microbiological, and sensory qualities depends on the chosen plants and their physical, chemical, and biological profiles.

### 3.2. Antioxidant Capacity

The results of the investigated phenolic and flavonoid contents of the kombucha samples and their controls are presented in Table 3. It can be observed that the fermented kombucha samples show significantly increased phenolic content compared to their non–fermented controls, especially for Basil and Hyssop batch C, indicating the highest phenolic activity (55.57 and 56.46 mg GA/100 mL). Compared to the literature, Villarreal-Soto et al. [35] used different extracts of black tea kombucha, and their results show that the content of total polyphenols grew only slightly after kombucha fermentation, from 3 to 8.9 and 215 to 221.6 mg GAE/100 mL.

**Table 3.** Phenols content (mg GA/100 mL) and flavonoid content (mg RU/100 mL).

| Kombucha Samples | Basil Batch C | Chamomile Batch A | Chamomile Batch B | Chamomile Batch C | Rosemary Batch C | Lavender Batch B | Lavender Batch C | Hyssop Batch B | Hyssop Batch C |
|---|---|---|---|---|---|---|---|---|---|
| Phenols | 55.57 ± 0.89 [c] | 12.65 ± 0.55 [a] | 12.40 ± 0.1 [a] | 42.11 ± 0.65 [c] | 45.57 ± 0.44 [c] | 24.37± 0.52 [b] | 51.58 ± 1.73 | 16.48 ± 1.0 [a] | 56.46 ± 0.44 [c] |
| Flavonoids | 0.25± 0.11 [a] | 0.67± 0.06 [a] | 1.04 ± 0.1 [b] | 5.07 ± 0.25 [c] | 1.07 ± 0.14 [b] | 2.58 ± 0.1 [b] | 2.93 ± 0.4 [b] | 0.84 ± 0.11 [a] | 0.49 ± 0.09 [a] |
| **Controls** | Basil batch C | Chamomile batch A | Chamomile batch B | Chamomile batch C | Rosemary batch C | Lavender batch B | Lavender batch C | Hyssop batch B | Hyssop batch C |
| Phenols | 20.36 ± 0.29 [b] | 2.26 ± 0.24 [a] | 4.60 ± 0.04 [a] | 31.32 ± 0.55 [c] | 41.52 ± 0.48 [c] | 4.37 ± 0.19 [a] | 23.02 ± 0.0 [b] | 1.85 ± 0.06 [a] | 11.37 ± 0.49 [b] |
| Flavonoids | 6.99 ± 0.17 [b] | 0.66 ± 0.15 [a] | 2.41 ± 0.36 [b] | 0.06 ± 0.02 [a] | 9.31 ± 0.13 | 2.84 ± 0.25 [b] | 3.41 ± 0.53 [b] | 0.56 ± 0.1 [a] | 0.96 ± 0.4 [a] |

Note: Different letters in columns indicate that there is a significant difference at $p \leq 0.05$.

The occurrence of increased phenolic compounds after fermentation is considered to be caused by the breakage of complex molecules after the release of enzymes from bacteria and yeasts [36]. Flavonoid content seems to decrease or stay the same during the fermentation process, except in Chamomile batch C, which has 5.07 mg RU/100 mL as opposed to its control of 0.06 mg RU/100 mL. The observed lowered flavonoids found in the kombucha samples match the results of Jakubczyk et al. [37], who obtained functional beverages from white, green, black, and red tea. The total flavonoids found in the different teas respectively consisted of 25.41, 23.17, 20.93, and 39.59 mg RU/100 mL, and after 14-day long fermentation, their flavonoid contents changed to 18.13, 12.67, 11.16, and 24.25 mg RU/100 mL. Similarly, the total phenolic content increases during the fermentation as the examples of the green and black tea kombucha show that the phenols content increases from 16.9 and 18.31 to 32.01 and 20.6 mg RU/100 mL, respectively.

Antioxidant capacity was analyzed using three antioxidant assays. The results show that both the kombucha samples and their controls demonstrate substantial activity (Table 4). Overall, of kombucha samples, Chamomile, Rosemary, and Lavender batch C reach the strongest antioxidant activity, and as for the controls, the Rosemary stands out the most.

**Table 4.** Antioxidant capacity (mmol TE/100 mL).

| Kombucha Samples | Basil Batch C | Chamomile Batch A | Chamomile Batch B | Chamomile Batch C | Rosemary Batch C | Lavender Batch B | Lavender Batch C | Hyssop Batch B | Hyssop Batch C |
|---|---|---|---|---|---|---|---|---|---|
| DPPH | 67.46 ± 1.91 | 41.30 ± 1.86 | 30.28 ± 1.02 | 64.25 ± 1.21 | 73.70 ± 4.84 | 60.41 ± 2.96 | 65.10 ± 0.61 | 34.51 ± 0.22 | 59.72 ± 2.90 |
| ABTS | 513.20 ± 14.49 | 264.11 ± 13.12 | 131.17 ± 1.37 | 615.62 ± 16.05 | 683.29 ± 16.80 | 395.46 ± 16.36 | 631.30 ± 18.11 | 192.25 ± 7.93 | 533.08 ± 38.48 |
| RP | 51.68 ± 0.79 | 19.45 ± 0.98 | 19.37 ± 0.28 | 65.27 ± 0.99 | 63.34 ± 2.68 | 31.20 ± 0.39 | 72.92 ± 1.12 | 27.21 ± 0.92 | 82.76 ± 0.25 |
| **Controls** | Basil batch C | Chamomile batch A | Chamomile batch B | Chamomile batch C | Rosemary batch C | Lavender batch B | Lavender batch C | Hyssop batch B | Hyssop batch C |
| DPPH | 54.98 ± 0.2 [d] | 6.66 ± 0.58 [a] | 13.74 ± 0.48 [b] | 56.09 ± 0.68 [d] | 70.87 ± 1.93 [d] | 14.14 ± 6.19 [b] | 50.96 ± 2.86 [d] | 7.71 ± 1.31 [a] | 33.04 ± 0.34 [c] |
| ABTS | 430.84 ± 17.8 [c] | 140.53 ± 2.62 [b] | 62.80 ± 8.37 [b] | 561.57 ± 75.89 [c] | 548.10 ± 5.5 [c] | 94.74 ± 12.05 [b] | 477.52 ± 7.99 [c] | 36.57 ± 4.5 [a] | 192.03 ± 28.61 [c] |
| RP | 50.45 ± 1.53 [c] | 14.88 ± 0.82 [a] | 8.72 ± 0.68 [a] | 50.52 ± 5.31 [c] | 66.13 ± 1.46 [c] | 12.89 ± 0.07 [a] | 69.90 ± 0.07 [c] | 3.59 ± 0.28 [a] | 34.07 ± 0.91 [b] |

DPPH—2,2-diphenyl-1-picrylhydrazyl; ABTS—2,2′-azino-bis-3-ethylbenzothiazoline-6-sulphonic acid; RP—reducing power (RP). Note: Different letters in columns indicate that there is a significant difference at $p \leq 0.05$.

While differences in the antioxidant capacity of each kombucha sample and their controls vary, kombucha fermentation is shown to have a positive effect in summary. Out of the performed antioxidant tests, the reducing power assay shows little to no change from the controls to the samples, while the ABTS shows the highest increase in activity. In comparison, different kombucha beverages made using by-products, such as coffee-fortified kombucha, are reported to have increased free radical scavenging activity of DPPH during fermentation, with ICP ranging from 26.13 to 170.23 µL/mL [38]. Also, research regarding kombucha-like beverages from acerola by-product (3% and 5%) showcased an increase in antioxidant activity (DPPH) for 14.8% (5.12 mmol/L) and 6.6% (5.98 mmol/L), respectively [39]. Shrirari and Satyanarayana [40] investigated changes in the free radical activity of kombucha during fermentation using an ABTS assay. Their results also showed growth in the antioxidant activity from a tea control of 43.3% to an 18-day-long fermented kombucha sample of 70.11%. This increase in the antioxidant activity of kombucha is claimed to be dependent on numerous factors, like the type of substrate, the fermentation time, and kombucha microbiota [39].

*3.3. Pharmacological Activities*

The results of the anti-inflammatory (AIA) and antihyperglycemic (AHgA) assays performed on the kombucha samples, as well as their controls, are shown in Table 5. They indicate positive pharmacological activity for all samples. The strongest AIA activity is found in the Basil and Hyssop samples (58.71 and 61.53), and the control in the Rosemary batch is the only one that had high activity before fermentation and kept the same percentages after it. Furthermore, the rest of the samples show a significant increase in AIA activity, up to double percentages. Kombucha's anti-inflammatory properties are strongly associated with the phenolic compounds produced during the fermentation process [36].

**Table 5.** Pharmacological activities (%).

| Kombucha Samples | Basil Batch C | Chamomile Batch A | Chamomile Batch B | Chamomile Batch C | Rosemary Batch C | Lavander Batch B | Lavander Batch C | Hyssop Batch B | Hyssop Batch C |
|---|---|---|---|---|---|---|---|---|---|
| AIA | 58.71 ± 0.93 b | 36.98 ± 1.12 a | 31.4 ± 1.08 a | 51.29 ± 0.96 b | 47.25 ± 2.56 b | 30.35 ± 1.04 a | 55.64 ± 3.07 b | 33.25 ± 0.28 a | 61.53 ± 0.59 b |
| AHgA | 25.13 ± 0.08 b | 14.52 ± 0.05 a | 11.64 ± 0.04 a | 26.89 ± 0.4 b | 25.74 ± 0.36 b | 13.98 ± 0.74 a | 29.74 ± 1.02 b | 11.53 ± 0.08 a | 30.69 ± 0.45 b |
| Controls | Basil batch C | Chamomile batch A | Chamomile batch B | Chamomile batch C | Rosemary batch C | Lavander batch B | Lavander batch C | Hyssop batch B | Hyssop batch C |
| AIA | 39.55 ± 2.64 c | 15.43 ± 0.06 a | 17.11 ± 0.12 a | 43.12 ± 3.11 c | 45.17 ± 3.88 c | 12.37 ± 0.09 a | 27.17 ± 0.05 b | 14.83 ± 0.03 a | 29.44 ± 1.65 b |
| AHgA | 17.11 ± 0.03 b | nd | nd | 21.36 ± 0.02 c | 23.12 ± 0.98 c | nd | 10.08 ± 0.09 c | nd | 10.95 ± 0.03 c |

AIA—anti-inflammatory activity; AHgA—antihyperglycemic assay. Note: Different letters in columns indicate that there is a significant difference at $p \leq 0.05$.

In the research of Villarreal-Soto et al. [35], non-fermented tea obtained 66% or even 0% inhibition compared to different kombucha extracts with 87–91% inhibition after 21-day fermentation, which led to the conclusion that kombucha extracts could potentially become an alternative for the development of non-steroidal drugs. Regarding the antihyperglycemic properties of the investigated kombucha samples, the results show noteworthy activity (Table 5), especially for Lavender and Hyssop batch C (29.74 and 30.69%). Some samples, like Chamomile and Rosemary batch C, present with high AHgA activity within their controls, which did not grow much after the fermentation, while others like Chamomile batches A and B, Lavender batch B, and Hyssop batch B do not show any AHgA activity before their kombucha fermentation. Regular kombucha is recorded to have antihyperglycemic activity based on various studies [41,42], but there is no direct comparison to the use of the same method found in the literature.

*3.4. Sensory Analysis*

Generally, there are no specific standards or regulations when it comes to the sensory analysis of kombucha beverages. The main reason for this problem lies in the infinity of the combination of the microbial consortium, the multiple processes used at home or in the industry, as well as the physico-chemical composition of the used material for kombucha preparation [33]. The diversity in sensory analysis among the authors is also reflected in the sensory panel concept. Namely, some studies are conducted with a trained sensory panel [33,43], but some studies include semi-trained or non-qualified sensory panels [44] to determine the sensory analysis of kombucha beverages; thus, the obtained results are significantly diverse among each other. The need for improved and standardized sensory research is overwhelming. In this way, deeper investigations of the olfactive and gustative dimensions of kombucha would be achieved.

Considering all the above, in this study, sensory characteristics and overall acceptance of prepared kombucha beverages were conducted by a trained sensory panel coupled with advanced mathematical modelling. A trained sensory panel was employed following a recommendation by Mintel [31], which indicated that consumers of kombucha beverages are mostly young females. The panel involved six sensory descriptors (smell, taste, acidity, sweetness, color tone, and overall acceptability). Descriptive sensory and ANOVA calculation results are summarized in Table 6. ANOVA analysis was performed for each descriptor (smell, taste, acidity, sweetness, color tone, and overall acceptability) to identify the descriptors with no product effect. According to the results, all selected descriptors are statistically significant (at level $p \leq 0.001$) in discriminating the samples, meaning that all of them are useful in characterizing the differences among products.

**Table 6.** Descriptive statistics table of sensory panel analysis.

| Products | Smell | Taste | Acidity | Sweetness | Color Tone | Overall Acceptability |
|---|---|---|---|---|---|---|
| Basil batch C | 4.10 ± 0.31 [c] | 4.30 ± 0.47 [d] | 2.35 ± 0.49 [b] | 3.45 ± 0.51 [d] | 2.45 ± 0.51 [b] | 4.70 ± 0.47 [d] |
| Chamomile batch A | 1.45 ± 0.51 [a] | 1.45 ± 0.51 [a] | 2.45 ± 0.51 [b] | 2.45 ± 0.51 [b] | 1.55 ± 0.51 [a] | 4.45 ± 0.51 [cd] |
| Chamomile batch B | 3.35 ± 0.49 [b] | 1.65 ± 0.49 [a] | 3.45 ± 0.51 [c] | 1.65 ± 0.49 [a] | 2.45 ± 0.51 [b] | 3.70 ± 0.47 [b] |
| Chamomile batch C | 1.65 ± 0.49 [a] | 3.45 ± 0.51 [c] | 3.45 ± 0.51 [c] | 1.30 ± 0.47 [a] | 3.65 ± 0.49 [c] | 3.35 ± 0.49 [b] |
| Lavender batch B | 4.55 ± 0.51 [c] | 4.20 ± 0.41 [d] | 1.35 ± 0.49 [a] | 3.40 ± 0.50 [d] | 2.60 ± 0.50 [b] | 4.75 ± 0.44 [d] |
| Lavender batch C | 4.45 ± 0.51 [c] | 1.50 ± 0.51 [a] | 3.55 ± 0.51 [c] | 1.50 ± 0.51 [a] | 4.60 ± 0.50 [d] | 3.50 ± 0.51 [b] |
| Hyssop batch B | 4.25 ± 0.44 [c] | 4.20 ± 0.41 [d] | 1.55 ± 0.51 [a] | 3.50 ± 0.51 [d] | 1.55 ± 0.51 [a] | 4.60 ± 0.50 [cd] |
| Hyssop batch C | 4.25 ± 0.44 [c] | 3.60 ± 0.50 [c] | 1.80 ± 0.41 [a] | 2.75 ± 0.44 [b] | 2.55 ± 0.51 [b] | 4.20 ± 0.41 [c] |
| Rosemary batch C | 4.25 ± 0.44 [c] | 2.25 ± 0.44 [b] | 4.75 ± 0.44 [d] | 1.20 ± 0.41 [a] | 2.30 ± 0.47 [b] | 1.50 ± 0.51 [a] |
| F test | 134.368 | 134.587 | 106.584 | 78.751 | 73.746 | 91.580 |
| *p*-value | $p \leq 0.001$ | $p \leq 0.001$ | $p \leq 0.001$ | $p \leq 0.001$ | $p \leq 0.001$ | $p \leq 0.001$ |

Note: Different letters in columns indicate that there is a significant difference at $p \leq 0.05$, according to Tukey's HSD test.

According to the results given in Table 6, vinegar notes are prominent in the Chamomile batches A and C, yeast smell is prominent in Chamomile batch B, while plant smell is characteristic for other kombucha samples. The taste profiles show that the vinegar taste (mild or strong) is in Chamomile batches A and B, Lavender batch C, and Rosemary batch C. In general, kombucha only starts to taste vinegar-like when it has been fermenting for too long or when the required fermentation conditions are not met. These also affected the ratings in acidity, sweetness, and overall acceptability. Rosemary batch C is highlighted as a too-sour drink, while Chamomile batches A and B and Lavender batch C are characterized as medium/noticeably sour. Further, the sweetness of these samples is correlated with acidity, so they are rated as imperceptibly or slightly sweet drinks. It can be explained by

the fact that during long fermentation, the yeast utilizes sugars and tannins in kombucha, transforming them into ethanol. The bacteria can use this ethanol, turning it into acidity and giving kombucha its distinctively sour taste [45].

From the proposed palette of kombucha drinks, the best performance or completely acceptable description is noted for Lavender batch B, Basil batch C, and Hyssop batch B. Chamomile batch A and Hyssop batch C are characterized as partially acceptable.

During the panel analysis, the following model was used: $Y = P + A + S + P \cdot A$, introducing the effects of products ($P$), assessors ($A$), and sessions ($S$) on sensory descriptors ($Y$). All factors were treated as random factors. The ANOVA table for the panel analysis is presented in Table 7. All descriptors allowed discriminating abilities of the investigated products, and they could be taken into account in the next steps of the analysis.

**Table 7.** ANOVA table of panel analysis parameters (type III sum of squares analysis).

| Source | Type | df | Smell | Taste | Acidity | Sweetness | Color Tone | Overall Acceptance |
|---|---|---|---|---|---|---|---|---|
| Products | Fixed | 8 | 232.578 ** | 243.044 ** | 203.444 ** | 148.844 ** | 148.700 ** | 169.878 ** |
| Assessors | Random | 9 | 1.800 | 4.756 | 3.689 | 0.356 | 15.800 ** | 2.583 |
| Sessions | Random | 1 | 0.356 | 0.200 | 0.200 | 0.022 | 0.022 | 0.006 |
| Products × Assessors | Random | 72 | 21.200 ** | 22.844 ** | 21.111 ** | 26.044 ** | 15.300 * | 24.567 ** |
| Error | | 89 | 13.644 | 10.800 | 15.800 | 13.978 | 11.978 | 12.494 |

** Statistically significant at $p < 0.001$ level; * statistically significant at $p < 0.05$ level.

In Table 8, ANOVA analysis was performed for each assessor, and for each product, to check if the descriptor were recognized by each assessor. The *p*-values displayed in Table 6 correspond to descriptors for which the assessor was able to differentiate the difference among the products. According to Table 8, assessors 1–9 recognize a smell between products, assessors 1–3 recognize the taste, while acidity is detected by all assessors. The results presented in Table 8 also show that assessors 2–9 recognize the sweetness between products, assessors 1 and 3–9 recognize the color tone, while overall acceptability is detected by assessors 2–10.

**Table 8.** ANOVA table of panel analysis parameters (type III sum of squares analysis).

| Assessors | Smell | Taste | Acidity | Sweetness | Color Tone | Overall Acceptability |
|---|---|---|---|---|---|---|
| 1 | 0.0003 | <0.0001 | <0.0001 | / | 0.0003 | / |
| 2 | 0.0001 | 0.0001 | 0.0002 | <0.0001 | / | 0.0001 |
| 3 | 0.0003 | 0.0001 | 0.0016 | 0.0049 | 0.0014 | 0.0001 |
| 4 | 0.0001 | / | 0.0062 | 0.0214 | 0.0028 | 0.0044 |
| 5 | 0.0012 | 0.0001 | 0.0001 | 0.0019 | 0.0002 | 0.0049 |
| 6 | <0.0001 | 0.0014 | 0.0001 | 0.0028 | <0.0001 | <0.0001 |
| 7 | 0.0007 | 0.0000 | 0.0070 | 0.0003 | 0.0005 | 0.0005 |
| 8 | 0.0006 | 0.0005 | 0.0002 | <0.0001 | 0.0007 | 0.0053 |
| 9 | 0.0000 | 0.0000 | 0.0003 | 0.0001 | 0.0001 | <0.0001 |
| 10 | / | / | <0.0001 | / | / | <0.0001 |

The results of the panel analysis are presented in Figure 2a–i, where the sensory scores of each product and the assessors for a set of descriptors are presented to compare product quality (averaged over two repetitions) and each assessor's sensory scores (averaged over the two repetitions) for the set of descriptors. Also, the thick red line corresponding to the average of all assessor's sensory scores is plotted in Figure 2a–i to compare the assessor's sensory scores to the average rating for each product for a set of descriptors.

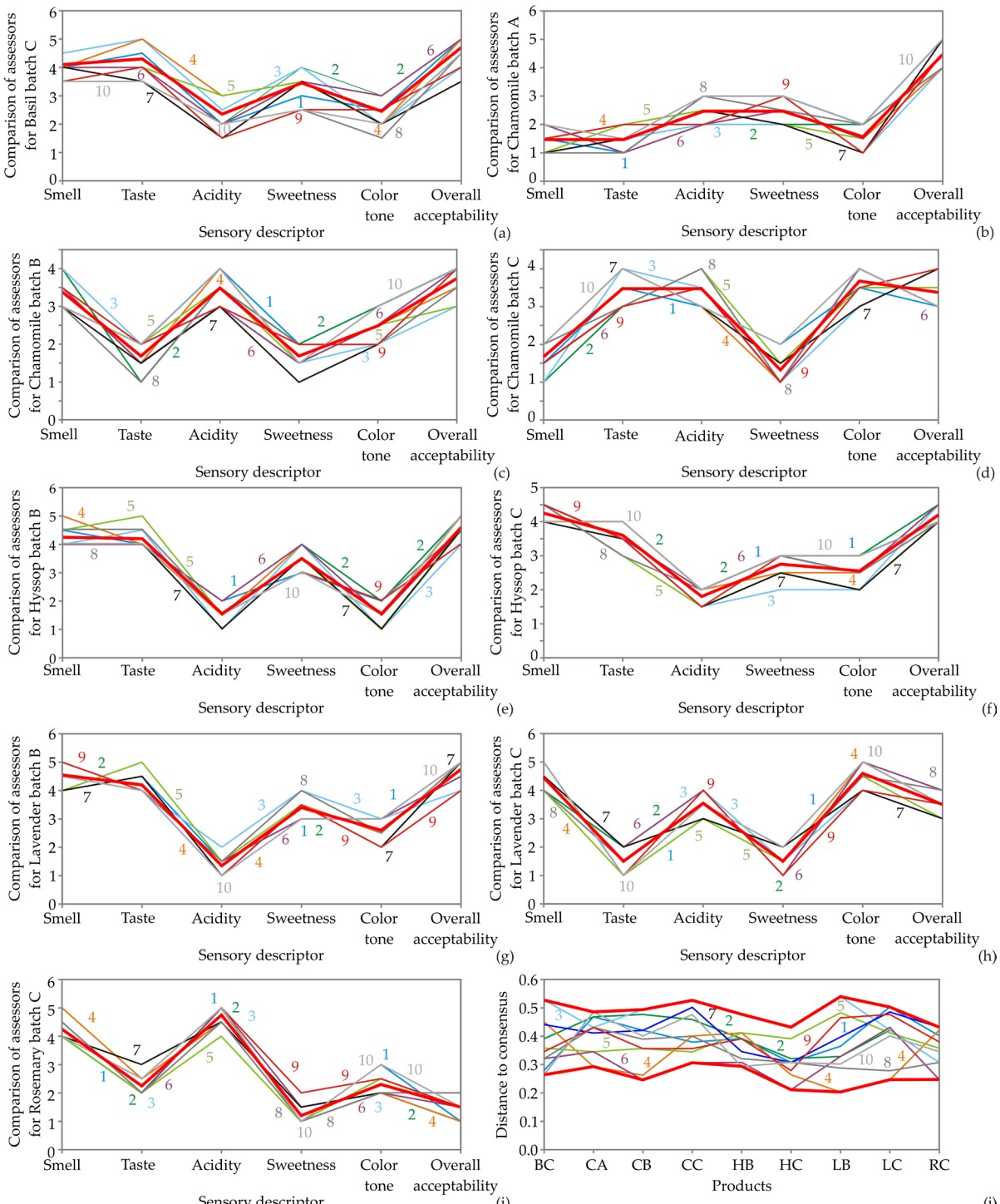

**Figure 2.** Sensory scores for products and assessors for a set of descriptors for (**a**) Basil batch B, (**b**) Chamomile batch A; (**c**) Chamomile batch B; (**d**) Chamomile batch C; (**e**) Hyssop batch B, (**f**) Hyssop batch C, (**g**) Lavender batch B, (**h**) Lavender batch C, (**i**) Rosemary batch C, and (**j**) Euclidean distance of the assessor's evaluation to consensus.

Furthermore, the Euclidean distance was estimated for each assessor's sensory evaluation for each product to the average score for all assessors and all descriptors. Figure 2j shows these distances for each product for all assessors, enabling the identification of the assessor's sensory score distance from the consensus. The lower the Euclidean distance,

the closer the assessor to the consensus. According to the results, assessors 3, 5, and 7 exert the highest distances to the consensus for product evaluation.

According to hierarchical clustering analysis (using Euclidean distances and complete distances) and based on the assessor's sensory scores of products evaluated by the sensory descriptors, four groups of assessors are recognized (Figure 3a). The first cluster comprises assessors 3, 4, and 7, while assessors 2 and 6 are in the second group, assessors 1 and 9 are in the third group, and assessors 5, 8, and 10 are in the fourth cluster. Each cluster contains assessors with similar sensory ratings of the products estimated by the sensory descriptors.

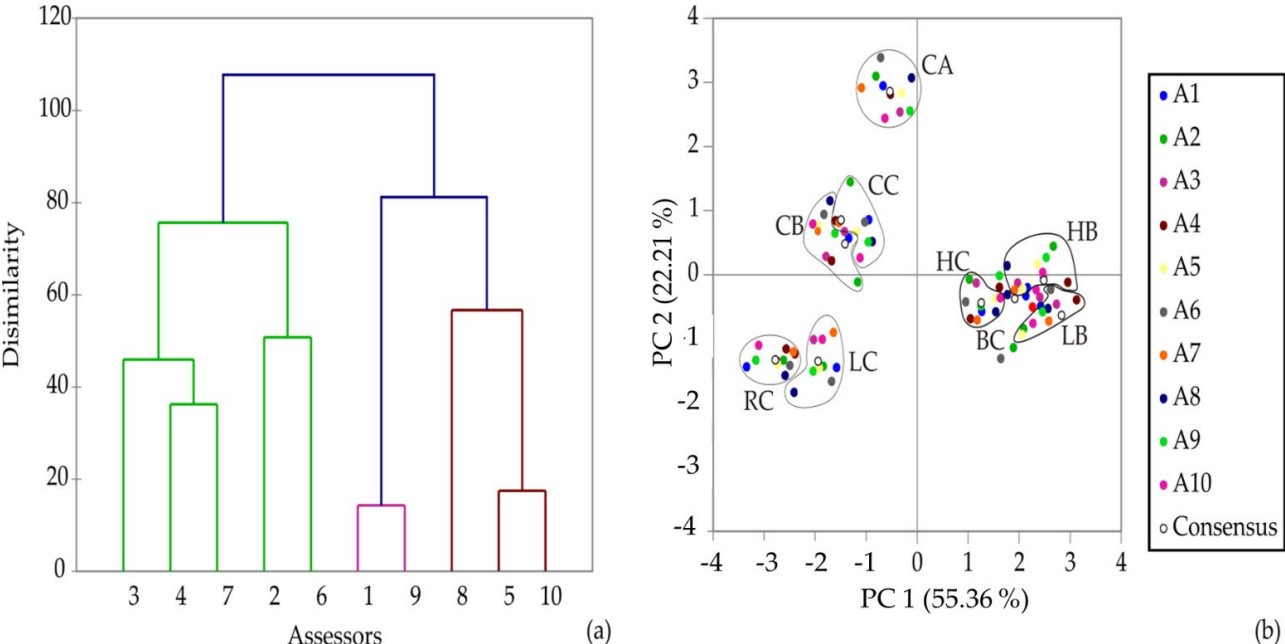

**Figure 3.** Multivariate analysis to discriminate the assessors' sensory scores of products evaluated by sensory descriptors: (**a**) hierarchical clustering of the panel analysis and (**b**) PCA ordination of the GPA results.

Generalized Procrustes analysis (GPA) was also used to reduce the scale of the effects of the product, the assessors, and the descriptor and to obtain a consensus configuration. Within this analysis, the proximity of sensory evaluation is presented between the descriptors used by different assessors to describe products. Principal component analysis (PCA) of the presented data explain that the first two components account for 77.57% of the total variance (55.36% and 22.21%, respectively). The results are divided into clusters, and the position of the sensory evaluations in the first-factor plane coincide well with the consensus configuration for all assessors (shown in Figure 3b).

## 4. Conclusions

According to the results in this study, it can be concluded that by-products obtained during the process of essential oil distillation (solid waste residue, wastewater, and hydrolate) from basil, chamomile, lavender, rosemary, and hyssop can be successfully utilized as cultivation media for kombucha fermentation. Based on the fermentation parameters, the most complete fermentation is obtained in the case of kombucha based on chamomile, while the less successful fermentation is observed for kombucha based on basil waste materials. Also, the presented kinetic can be used as a tool for monitoring the kombucha fermentation and tracking the defined fermentation parameters while obtaining the final product. The final kombucha beverages prepared on the alternative substrates have a potential regarding their phenolic content, antioxidant capacity, and pharmacological activities, since a positive trend for all the kombucha samples is observed compared to the control samples. The undertaken sensory analysis indicates that the complete acceptability is determined for

kombucha beverages originating from lavender and hyssops (when they are made from solid waste stream mixed with hydrolate) as well as basil (when the kombucha is made from concentrated wastewater and hydrolate). The promising results presented in this study are pioneer results regarding using the complete waste material from the essential oil distillation process to obtain kombucha, but further investigation needs to focus on comprehensive chemical profiling, the antimicrobial potential, and the determination functionality of the gained beverages before optimization and commercialization steps.

**Author Contributions:** Conceptualization, A.R., O.Š. and A.T.; methodology, A.R.; validation, L.P., O.Š. and A.S.; formal analysis, V.T., M.A. and J.V.; investigation, D.C. and G.Ć.; resources, M.A. and A.R.; data curation, L.P. and O.Š.; writing—original draft preparation, O.Š.; writing—review and editing, A.T., L.P. and O.Š.; visualization, L.P.; supervision, S.M. All authors have read and agreed to the published version of the manuscript.

**Funding:** The Ministry of Education, Science and Technological Development of the Republic of Serbia (Grant no: 451-03-68/2022-14/200032 and 451-03-68/2022-14/200134) was financially supported this publication.

**Institutional Review Board Statement:** Not applicable.

**Informed Consent Statement:** Not applicable.

**Data Availability Statement:** Not applicable.

**Conflicts of Interest:** The authors declare no conflict of interest.

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
