# Peer review of "Biological Potential of Alternative Kombucha Beverages Fermented on Essential Oil Distillation By-Products"

_fermentation, doi:10.3390/fermentation8110625_

Round 1
Reviewer 1 Report
In general, a very interesting piece of work with Kombucha beverages design to add value to byproducts, which is a trending topic in the agro-food sciences area.
Minor comments on data presentation and discussion of results.
Section 2.3. phytochemicals. Not correct, there is not any phytochemical analysis (minimum HPLC analysis) for the qualitative and quantitative evaluation of the products and the bevereges. Therefore, the information about Folin (TPCs) or colorimetric assay of flavonols (TFCs) can be incorporated to the section of antioxidant capacity tests and assays, that can be now section 2.3., all together.
Total polyphenols are usually expressed as gallic acid equivalents (GAE), but other phenolic compounds such as catechin/epicatechin or caffeic acid have also been used for standardisation. This standardisation refers to the traditional spectrophotometrical measurement of total polyphenols using the Folin-Ciocalteau method (Singleton and Rossi, 1965), which is based on reducing capacity. The method is not specific for polyphenols because other reducing compounds such as ascorbic acid, sugars and proteins will also be included in the quantification, thus leading to an overestimation of the actual polyphenol content. The total polyphenol content assessed with this method is not suitable for characterisation of polyphenols in foods.
2.5. Pharmacological activities, it would be desirable to compare data with IC50 value of a known drug as positive control either as antinflammatory and/or antihyperglycemic. In this way, through the discussion, it would be possible to make comparisons for the potential interest of these beverages n this sense, as modulators of inflammation or potential hypoglycemic products. The use of % as units not as desirable as units/mL of drink, in order to compare with other products.
3.2. Phytochemical composition. Delete the header. Use the data of TPCs and Total flavonoid contents as colorimetric assays together with the antioxidant capacity tests.
Table 3 "Total phenolic contents" and "total flavonoid contents" to be commented and discussed with the antioxidant capacity tests.
Do not use antioxidant "activity", term. The "activity" refers to pure compound, not to a mix or a complex mix of compounds as present in the beverages. The correct form is "antioxidant capacity" tests/assays.
4. Conclusions.
The ln. 557-558, is speculative.
The kombucha beverages prepared by using alternative substrates showed ......antioxidant capacity... etc.
There is not any data presented on the phenolic content/composition/identities of any specific compound to be connected with the specific result of bioactivity.
In order to use the kombucha drinks as "valuable functional beverages" a minimum characterisation of the starting products, and the composition of the drinks in terms of phenolic compounds (individual compounds identified and quantified), is needed.
The vague concepts of "functional product" without a good and complete characterisation of the product (minimum with a HPLC analysis) is not acceptable. This last sentence of the conclusion should be reformulated or eliminated.
Author Response
Reviewer 1
In general, a very interesting piece of work with Kombucha beverages design to add value to byproducts, which is a trending topic in the agro-food sciences area.
ANSWER: The Authors would like to thank the Reviewer for a quick and professional review as well as the opportunity to make essential and crucial changes in our work. The Reviewer' remarks are accepted and disscused. The paper is changed according to their comments. The Authors believe that the changed paper would satisfy the Reviewer' criteria and that it is going to be interesting enough for publishing in the Fermentation.
We decided to revise the manuscript according to the Reviewer' remarks, highlighting the changes directly in the revised manuscript.
Minor comments on data presentation and discussion of results.
Section 2.3. phytochemicals. Not correct, there is not any phytochemical analysis (minimum HPLC analysis) for the qualitative and quantitative evaluation of the products and the bevereges. Therefore, the information about Folin (TPCs) or colorimetric assay of flavonols (TFCs) can be incorporated to the section of antioxidant capacity tests and assays, that can be now section 2.3., all together.
ANSWER: Thank you for this suggestion, we replace this methodology in the following section and renamed as “Antioxidant capacity”. Also, we are conscious of the lack of HPLC analysis, but we do not have opportunity to use this technique for this study.
Total polyphenols are usually expressed as gallic acid equivalents (GAE), but other phenolic compounds such as catechin/epicatechin or caffeic acid have also been used for standardisation. This standardisation refers to the traditional spectrophotometrical measurement of total polyphenols using the Folin-Ciocalteau method (Singleton and Rossi, 1965), which is based on reducing capacity. The method is not specific for polyphenols because other reducing compounds such as ascorbic acid, sugars and proteins will also be included in the quantification, thus leading to an overestimation of the actual polyphenol content. The total polyphenol content assessed with this method is not suitable for characterisation of polyphenols in foods.
ANSWER: Thank you for this comment. We agree that the used methodology have disadvantages like any other. We think that the used methods were adequate for primary screening of the potential of our samples and comparation with controls (controls initially have sugars which utilisation during succesuful fermentation, while ascorbic acid and proteins are not characteristic for cultivation medium for kombucha), and further work will imply using a better methodology in view of definition of the (phyto)chemical compositions of kombucha beverages.
2.5. Pharmacological activities, it would be desirable to compare data with IC50 value of a known drug as positive control either as antinflammatory and/or antihyperglycemic. In this way, through the discussion, it would be possible to make comparisons for the potential interest of these beverages n this sense, as modulators of inflammation or potential hypoglycemic products. The use of % as units not as desirable as units/mL of drink, in order to compare with other products.
ANSWER: Thank you for this suggestion. As standard methods for antinflammatory and antihyperglycemic activity were involved in this study and implied comparing absorbances between reaction mixture with investigated samples and a control mixture without a sample, obtained values are presented in percentages (%). IC50 value shows concentration of compound needed for reaching 50% of antinflammatory/antihyperglycemic activity. To calculate IC 50 value, concentration of the bioactive compound would need to be known, which in this case is not possible as we are measuring activity of kombucha from waste with unknown compounds and their concentrations.
3.2. Phytochemical composition. Delete the header. Use the data of TPCs and Total flavonoid contents as colorimetric assays together with the antioxidant capacity tests. Table 3 "Total phenolic contents" and "total flavonoid contents" to be commented and discussed with the antioxidant capacity tests.
ANSWER: Thank you for this suggestion. We renamed the heading and connected the mentioned results.
Do not use antioxidant "activity", term. The "activity" refers to pure compound, not to a mix or a complex mix of compounds as present in the beverages. The correct form is "antioxidant capacity" tests/assays.
ANSWER: We agree with this comment; therefore, we renamed “antioxidant activity” to “antioxidant capacity” throughout the manuscript.
- Conclusions.
The ln. 557-558, is speculative. The kombucha beverages prepared by using alternative substrates showed ......antioxidant capacity... etc. There is not any data presented on the phenolic content/composition/identities of any specific compound to be connected with the specific result of bioactivity.
ANSWER: Thank you for these suggestions. We absolutely agree with You, so we rewrote this section for a better image of the obtained results and all advantages and disadvantages of the manuscript.
In order to use the kombucha drinks as "valuable functional beverages" a minimum characterisation of the starting products, and the composition of the drinks in terms of phenolic compounds (individual compounds identified and quantified), is needed.
ANSWER: Thank you for this suggestion, we added that further investigation implies complete chemical characterisation and determination of the functionality of our products.
The vague concepts of "functional product" without a good and complete characterisation of the product (minimum with a HPLC analysis) is not acceptable. This last sentence of the conclusion should be reformulated or eliminated.
ANSWER: Thank you for this comment. We decided to eliminate the last sentence and adopt a new version of the conclusion.

Reviewer 2 Report
I am very grateful you for the invitation to review the manuscript fermentation-2028090 by Ranitović and coauthors "Biological potential of alternative kombucha beverages fermented on essential oil distillation by-products”. This article investigated the biological profiling and sensory analysis of newly obtained kombucha beverages based on alternative fermentation substrates. The work is very interesting but needs adjustments to increase the quality of the material.
Comments:
- Abstract, Line 9: Indicate the role of essential oils in the Kombucha fermentation process.
- Abstract: Present the most specific results. Insert numerical results related to the main findings of the work.
- Introduction: It is important to highlight global essential oil production and the amount of waste generated along the chain, which can be used in processes such as the one proposed.
- Introduction: Indicate the role of essential oils in the Kombucha fermentation process.
- Lines 42-43: Cite the main related bioactivity.
- Line 47: Indicate the legal issues regarding the use of other components in the production of Kombucha, since different countries determine which ingredients can be applied in the name of the beverage.
- Lines 79-91: Check throughout the text to use italic formatting to plant species.
- Line 89: Completely describe the abbreviations in the sentence “to Eur. Ph. [18]”.
- Lines 98-99: Check throughout the text to use italic formatting to microorganism’s species.
- Line 100: It is not clear whether the “cultivation medium” uses tea or is made only with the residue obtained from the extraction of essential oil.
- Line 148: Check sentence punctuation.
- Lines 180-185: The theoretical reference must be in the introduction or in the discussion of the results.
- Standardize units throughout the text (e.g., lines 207; 210; 203;249).
- Line 233: Please indicate the temperature in the sentence “refrigerated before consummation”.
- Line 261-263: Indicate the legal issues regarding the use of other components in the production of Kombucha, since different countries determine which ingredients can be applied in the name of the beverage.
- Lines 274-279: The information has already been presented in the material and methods. Consider removal to make the text less repetitive.
- Figure 1: Review the resolution of Figure 1.
- Lines 301-302: Describe the verified biochemical process. Even though it is something to be expected, it is important to complement the information.
- Lines 347-373: Please review the sentences for clarity and consider reorganizing some information in the introduction, especially obtaining the residues.
- Discussion: Essential oils are generally associated with antimicrobial effects. How might this impact the Kombucha production process?
- 3.3. Please indicate how each of the methods can be correlated with the compounds present in the residues and their effects.
- Lines 448-460: This information is not required. The concept of using sensory analysis is widely known and therefore it is a tool usually applied for this purpose.
- Lines 483-485: Include reference citation.
- Figure 2: Review the resolution of Figure 2.
Author Response
Reviewer 2
I am very grateful you for the invitation to review the manuscript fermentation-2028090 by Ranitović and coauthors "Biological potential of alternative kombucha beverages fermented on essential oil distillation by-products”. This article investigated the biological profiling and sensory analysis of newly obtained kombucha beverages based on alternative fermentation substrates. The work is very interesting but needs adjustments to increase the quality of the material.
ANSWER: The Authors would like to thank the Reviewer for a quick and professional review as well as the opportunity to make essential and crucial changes in our work. All the Reviewer' remarks are accepted and discussed. The paper is changed according to their comments. The Authors believe that the changed paper would satisfy the Reviewer' criteria and that it is going to be interesting enough for publishing in the Fermentation.
We decided to revise the manuscript according to the Reviewer' remarks, highlighting the changes directly in the revised manuscript.
Comments:
- Abstract, Line 9: Indicate the role of essential oils in the Kombucha fermentation process.
ANSWER: Thank you for this suggestion, we changed the text and indicate the role of waste streams in essential oil production for obtaining the kombucha.
- Abstract: Present the most specific results. Insert numerical results related to the main findings of the work.
ANSWER: Thank you for this suggestion, we modified the Abstract completely.
- Introduction: It is important to highlight global essential oil production and the amount of waste generated along the chain, which can be used in processes such as the one proposed.
ANSWER: Thank you for this observation, we added this information at the end of the first paragraph in the Introduction.
- Introduction: Indicate the role of essential oils in the Kombucha fermentation process.
ANSWER: Thank you for this suggestion. To our knowledge, no one used essential oil production waste streams or essential oil to make kombucha or improve kombucha taste. We highlighted this in the aim.
- Lines 42-43: Cite the main related bioactivity.
ANSWER: Thank you for this suggestion, we added the three most mentioned through the scientific-relevant literature.
- Line 47: Indicate the legal issues regarding the use of other components in the production of Kombucha, since different countries determine which ingredients can be applied in the name of the beverage.
ANSWER: Thank you for this suggestion. The legal issues about kombucha are very problematic. A Codex Standard for kombucha does not currently exist. Some countries such as Uganda, the USA, and Brazil have some recommendations about this product which is included in part of the fermented food concept. Additionally, kombucha fermentation is categorized as a specialized process in the FDA Food Code, requiring all retail or food service operators planning to sell kombucha to obtain a variance from their regulatory authority and to submit a food safety plan to their regulatory authority as defined in the Food Code section 3-502.11. On the other hand, U.S. FDA regulation 21 CFR 101.22 defines and guides essential oils labeling. Section 182.20 list indicates that essential oils are defined as GRAS (generally recognized as safe), while waste stream residues in essential oil production did not mention in this document. Finally, we think that this part of the quality management system required special attention, so we excluded this part from our current manuscript. In the following research steps, where we want to define the functionality of the obtained kombucha beverages, we are going to do a comprehensive approach to legal possibilities to commercialize the product.
- Lines 79-91: Check throughout the text to use italic formatting to plant species.
ANSWER: Thank you for this observation, we made changes throughout the text.
- Line 89: Completely describe the abbreviations in the sentence “to Eur. Ph. [18]”.
ANSWER: Thank you for this observation, we changed it.
- Lines 98-99: Check throughout the text to use italic formatting to microorganism’s species.
ANSWER: Thank you for this observation, we changed it.
- Line 100: It is not clear whether the “cultivation medium” uses tea or is made only with the residue obtained from the extraction of essential oil.
ANSWER: Thank you for this observation, we rewrote the whole paragraph in order to better explain the formation of cultivation medium in all three batches.
- Line 148: Check sentence punctuation.
ANSWER: Thank you for this observation, we changed it.
- Lines 180-185: The theoretical reference must be in the introduction or in the discussion of the results.
ANSWER: Thank you for this comment, we agree, so we deleted this sentence from the materials and methods part.
- Standardize units throughout the text (e.g., lines 207; 210; 203;249).
ANSWER: Thank you for this observation; we uniformed all units throughout the text.
- Line 233: Please indicate the temperature in the sentence “refrigerated before consummation”.
ANSWER: Thank you for this suggestion, we added temperature value in the text.
- Line 261-263: Indicate the legal issues regarding the use of other components in the production of Kombucha, since different countries determine which ingredients can be applied in the name of the beverage.
ANSWER: Thank you for this suggestion. The legal issues about kombucha are very problematic. A Codex Standard for kombucha does not currently exist. Some countries such as Uganda, the USA, and Brazil have some recommendations about this product which is included in part of the fermented food concept. Additionally, kombucha is categorized as a specialized process in the FDA Food Code, requiring all retail or food service operators planning to sell kombucha to obtain a variance from their regulatory authority and to submit a food safety plan to their regulatory authority as defined in the Food Code section 3-502.11. On the other hand, U.S. FDA regulation 21 CFR 101.22 defines and guides essential oils labeling. Section 182.20 list indicates that essential oils are defined as GRAS (generally recognized as safe), while waste stream residues in essential oil production did not mention in this document. Finally, we think that this part of the quality management system required special attention, so we excluded this part from our current manuscript. In the following research steps, where we want to define the functionality of the obtained kombucha beverages, we are going to do a comprehensive approach to legal possibilities to commercialize the product.
- Lines 274-279: The information has already been presented in the material and methods. Consider removal to make the text less repetitive.
ANSWER: Thank you for this suggestion, we made the suggested change.
- Figure 1: Review the resolution of Figure 1.
ANSWER: Thank you for this suggestion, we added the image with a better resolution.
- Lines 301-302: Describe the verified biochemical process. Even though it is something to be expected, it is important to complement the information.
ANSWER: Thank you for this suggestion, we added additional information about followed parameters.
- Lines 347-373: Please review the sentences for clarity and consider reorganizing some information in the introduction, especially obtaining the residues.
ANSWER: Thank you for this suggestion. We rewrote this paragraph, and replace some sentences in the introduction, while the rest of the information was adjusted in the following text.
- Discussion: Essential oils are generally associated with antimicrobial effects. How might this impact the Kombucha production process?
ANSWER: Thank you for this comment. Most essential oils indeed have an antimicrobial effect. On the other hand, there is a lack of information about the antimicrobial potential for almost all essential oil waste streams. Also, kombucha beverages have the potential to inhibit some microorganisms since acid production is present. Therefore, we added in conclusion that further investigation needs to be directed in comprehensive profiling in view of the chemical composition as well as the antimicrobial potential of the obtained kombucha beverages. In this way, a complete image of the functionality can be achieved.
- 3.3. Please indicate how each of the methods can be correlated with the compounds present in the residues and their effects.
ANSWER: Without the HPLC analyses (we are aware of lacking this in our actual study) and certain knowledge of compounds present in kombucha samples, we can’t correlate any specific compound to antioxidant tests or discuss antioxidant activity in a better view. Based on the information found in the literature about used plants and their essential oils there was an assumption about compounds expected to be in fermented drinks, including some antioxidants. Also, we have experience with this type of sample, since some coauthors have worked with kombucha during the last two decades. Antioxidant assays were used to confirm this suspicion and further analyses need to be done in order to fully identify and confirm the residues responsible for antioxidant activity.
- Lines 448-460: This information is not required. The concept of using sensory analysis is widely known and therefore it is a tool usually applied for this purpose.
ANSWER: Thank you for this suggestion, we deleted the first sentence.
- Lines 483-485: Include reference citation.
ANSWER: Thank you for this comment, we added the reference.
- Figure 2: Review the resolution of Figure 2.
ANSWER: Thank you for this suggestion. We added the image in better quality.

Round 2
Reviewer 2 Report
After carefully checking the responses and the revisions, the manuscript is suggested to be accepted by Fermentation.